# Longitudinal Study of Advanced Non-Small Cell Lung Cancer with Initial Durable Clinical Benefit to Immunotherapy: Strategies for Anti-PD-1/PD-L1 Continuation beyond Progression

**DOI:** 10.3390/cancers15235587

**Published:** 2023-11-26

**Authors:** Ivan Pourmir, Reza Elaidi, Zineb Maaradji, Hortense De Saint Basile, Monivann Ung, Mohammed Ismaili, Laure Fournier, Bastien Rance, Laure Gibault, Rym Ben Dhiab, Benoit Gazeau, Elizabeth Fabre

**Affiliations:** 1Department of Thoracic Oncology, Georges Pompidou European Hospital, CARPEM, Paris Cité University, 20 Rue Leblanc, 75015 Paris, France; relaidi@gmail.com (R.E.); zineb.maaradji@aphp.fr (Z.M.); mohammed.ismaili@aphp.fr (M.I.); rym.ben-dhiab@aphp.fr (R.B.D.); benoit.gazeau@aphp.fr (B.G.); elizabeth.fabre@aphp.fr (E.F.); 2INSERM U970, Université Paris-Cité, 56 Rue Leblanc, 75015 Paris, France; 3Department of Radiology, Georges Pompidou European Hospital, CARPEM, AP-HP Paris Cité University, 20 Rue Leblanc, 75015 Paris, France; monivann.ung@aphp.fr (M.U.); laure.fournier@aphp.fr (L.F.); 4Department of Medical Bioinformatics, Georges Pompidou European Hospital, 20 Rue Leblanc, 75015 Paris, France; bastien.rance@aphp.fr; 5Department of Pathology, Georges Pompidou European Hospital, AP-HP, Cité University, 20 Rue Leblanc, 75015 Paris, France; laure.gibault@aphp.fr

**Keywords:** non-small cell lung cancer, immune checkpoint inhibitors, PD-1, acquired resistance, tumor growth rate, oligoprogression

## Abstract

**Simple Summary:**

Anti-PD-1/PD-L1 treatments still face the problem of resistant diseases in non-small cell lung cancer. This resistance can be classified as primary or secondary (acquired). The existing literature suggests different outcomes and mechanisms in both situations, yet they need to be comparatively characterized. This unique comparative study reveals diverging survival and progression patterns: initial durable clinical benefit (minimum of 6 months of controlled disease) prior to progression associated with longer post-progression survival. In this situation, progression most often occurs in pre-existing lesions and rarely spreads to other sites. The radiological measure of the tumor growth rate also appears as a useful tool to help decide anti-PD-1/PD-L1 continuation. Altogether, these results benefit our understanding and management of resistance to anti-PD-1/PD-L1.

**Abstract:**

Background and aim: A better understanding of resistance to checkpoint inhibitors is essential to define subsequent treatments in advanced non-small cell lung cancer. By characterizing clinical and radiological features of progression after anti-programmed death-1/programmed death ligand-1 (anti-PD-1/PD-L1), we aimed to define therapeutic strategies in patients with initial durable clinical benefit. Patients and methods: This monocentric, retrospective study included patients who presented progressive disease (PD) according to RECIST 1.1 criteria after anti-PD-1/PD-L1 monotherapy. Patients were classified into two groups, “primary resistance” and “Progressive Disease (PD) after Durable Clinical Benefit (DCB)”, according to the Society of Immunotherapy of Cancer classification. We compared the post-progression survival (PPS) of both groups and analyzed the patterns of progression. An exploratory analysis was performed using the tumor growth rate (TGR) to assess the global growth kinetics of cancer and the persistent benefit of immunotherapy beyond PD after DCB. Results: A total of 148 patients were included; 105 of them presented “primary resistance” and 43 “PD after DCB”. The median PPS was 5.2 months (95% CI: 2.6–6.5) for primary resistance (*p* < 0.0001) vs. 21.3 months (95% CI: 18.5–36.3) for “PD after DCB”, and the multivariable hazard ratio was 0.14 (95% CI: 0.07–0.30). The oligoprogression pattern was frequent in the “PD after DCB” group (76.7%) and occurred mostly in pre-existing lesions (72.1%). TGR deceleration suggested a persistent benefit of PD-1/PD-L1 blockade in 44.2% of cases. Conclusions: PD after DCB is an independent factor of longer post-progression survival with specific patterns that prompt to contemplate loco-regional treatments. TGR is a promising tool to assess the residual benefit of immunotherapy and justify the continuation of immunotherapy in addition to radiotherapy or surgery.

## 1. Introduction

### 1.1. Epidemiology of Non-Small Cell Lung Cancer

Lung cancer is currently the leading cause of cancer death in the world, with non-small cell lung cancer (NSCLC) being the most frequent subtype [1]. Localized NSCLC is a curable disease albeit a remaining risk of localized and/or distant recurrence after treatment, increasing with stage [2]. The advanced stages of the disease (aNSCLC) are associated with poor prognosis in historical cohorts with a median survival of around 3 months in untreated patients and 8 months under conventional chemotherapies [3].

### 1.2. Current Immunotherapies: Indications of Anti-PD-1/PD-L1 Monotherapy

Advances in cancer immunotherapy have significantly improved this situation. NSCLC is among the first cancers that benefited from immune checkpoint inhibitors (ICI), especially monoclonal antibodies blocking the programmed death-1/programmed death ligand-1 axis (anti-PD-1/PD-L1), with profound and durable responses in some cases [4,5]. These drugs have been first approved as monotherapy for aNSCLC in 2015 [6,7,8]. They were designed to block signaling pathways dampening the anti-tumor immune response, although their precise pharmacodynamics are still not comprehensively elucidated [9,10].

### 1.3. Definitions for Resistance to Anti-PD-1/PD-L1

Despite the successes of ICI, the majority of aNSCLC patients will present a primary or secondary resistance. Resistance to anti-PD-1/PD-L1 leads to new challenges in the understanding of the underlying biological processes and optimal therapeutic strategies. In order to standardize studies, the Society of Immunotherapy of Cancer (SITC) task force advised a classification for this outcome as follows [11]:–Primary resistance, if tumor progression per RECIST 1.1 occurs < 6 months after anti-PD-1/PD-L1 initiation;–Secondary resistance, if tumor progression occurs under anti-PD-1/PD-L1 treatment and after ≥6 months of clinical benefit from the start of the therapy;–Late progression, if tumor progression occurs after anti-PD-1/PD-L1 discontinuation and after ≥6 months of clinical benefit from the start of the therapy.

Secondary resistance and late progression are less often encountered than primary resistance [5,12,13].

### 1.4. Need to Characterize Progression after Durable Clinical Benefit (PD after DCB)

Durable clinical benefit (DCB) of ICI is defined by the SITC, using the RECIST 1.1 criteria, as a ≥6-month duration of controlled disease [11]. Our limited knowledge of PD after DCB is explained by the scarcity of clinical and biological data and the various definitions of resistance in the literature [11]. However, a better understanding of PD after DCB mechanisms is crucial to adapt the therapeutic strategy to the specificities of those patients [14].

Tian et al. recently reported an association between PFS under an anti-PD-1/PD-L1 regimen and post-progression survival (PPS) of aNSCLC, but these retrospective data were focused on patients treated beyond progression, and only 4% of the cohort was under monotherapy [15]. Treatment continuation beyond progression is increasingly attempted on the basis of the theoretical pharmacodynamics of immunotherapy, even in clinical trials [16]. Yet, there is a lack of harmonization of practices based on objective data in this new clinical setting.

In particular, it is essential to objectively assess the residual benefit of anti-PD-1/PD-L1, beyond progression. The tumor growth rate (TGR) appears as a promising tool to measure the effect of anti-PD-1/PD-L1 in NSCLC [17,18].

In order to define therapeutic strategies beyond anti-PD-1/PD-L1 resistance, we collected clinical and radiological data of aNSCLCs on anti-PD-1/PD-L1 monotherapy treated in our center. Monotherapy was chosen in order to investigate the resistance phenomenon to a pure PD-1/PD-L1 blockade. Indeed, combinations such as chemo-immunotherapy or anti-PD-1/anti-CTLA-4 preclude robust conclusions on anti-PD-1/PD-L1 escape.

This enabled us, based on the SITC guidelines, to dichotomize cases of “PD after DCB and “primary resistance”. We then performed a comparative analysis of post-progression survival (PPS) between these groups and an in-depth characterization of “PD after DCB” group.

An exploratory analysis was performed using TGR to assess the global growth kinetics of cancer and the remaining benefit of immunotherapy beyond PD after DCB.

## 2. Materials and Methods

### 2.1. Cohort Selection, Exclusion Criteria, Radiological Definitions

This monocentric, retrospective study included patients from the European Georges Pompidou Hospital who started anti-PD-1/PD-L1 monotherapy between 30 June 2015 and 31 October 2019 and presented progressive disease (PD) documented according to RECIST 1.1 criteria after at least 1 cycle of anti-PD-1/PD-L1

Patients were excluded in case of another active neoplasm in the last 5 years, prior anti-PD-1/PD-L1 treatment, concomitant immunosuppressive therapy, non-cancer related death before PD, and insufficient baseline imaging data for subsequent disease assessment. Included patients were classified into 2 groups:–Anti-PD-1/PD-L1 “primary resistance” if PD occurred less than <6 months after anti-PD-1/PD-L1 initiation;–“Progression after DCB” if PD occurred at least ≥6 months after anti-PD-1/PD-L1 initiation.

Patients of the “PD after DCB” group were either categorized as oligoprogressive (OPD), if the number of progressive lesions was ≤3 (including intracranial lesions), or non-OPD in case of >3 progressive lesions [19,20].

The present study has been accepted and registered to the relevant institutional research and ethical committee (IRB registration #00011928, 12 October 2020, CERAPHP Centre). It has been conducted in accordance with the Declaration of Helsinki. All patients provided informed consent under the European Georges Pompidou Hospital-approved protocol allowing collection and analysis of data.

### 2.2. Clinical, Pathological and Radiological Data

Clinical, pathological, and radiological data were extracted from electronic patient files. PD-L1 expression prior to anti-PD-1/PD-L1 treatment was quantified by a specialized pathologist using PD-L1 immunohistochemical staining of tumor samples (antibody: PharmDx 22C3, Agilent) and expressed as percentage of positive tumor cells (Tumor Proportion Score—TPS). TPS resulted from exploratory analyses run on a subset of patients before 1st-line Pembrolizumab approval and, after that, from routine pathological evaluation, as required by European and French guidelines.

Follow-up included contrast-enhanced chest–abdominal–pelvic computer tomography (CT) scans and brain MRI when warranted. RECIST1.1 assessment was collected from imaging reports for all patients. For the “PD after DCB” group, RECIST1.1 evaluation, OPD status assessment, and measures were performed by a radiologist specialized in lung cancer immunotherapy (Dr. M. Ung) and a medical oncologist trained in radiological evaluation (Dr. I. Pourmir). The consecutive imaging performed after the first PD was also systematically reviewed for this cohort.

Survival data were completed from the publicly available national death register (Institut National de la Statistique et des Etudes Economiques). Study data were collected and managed using REDCap electronic data capture tools hosted at our institution [21,22].

### 2.3. Tumor Kinetics Evaluation by Tumor Growth Rate (TGR)

The global growth kinetics of cancer was estimated using the TGR, as previously published [17,18]. This method allows us to estimate the percentage of tumor burden increase per month. For this purpose, the global tumor burden of a given patient is modeled as a virtual tumor sphere growing exponentially over time, with its volume *V_t_* given at time *t* in the formula *V_t_* = *V*_0_.*exp*(*TG.t*). The diameter of this virtual tumor sphere equals the sum of longest diameters of target lesions via RECIST 1.1. Thus, having the diameter at two different time points, the parameter “TG” can be retrieved from the formula (Appendix A).

TGR was assessed for the “PD after DCB” group at ICI start and at progression, provided there was no change in antineoplastic treatments between the 2 consecutive computer tomography (CT) scans. The TGR was also used to specifically estimate the growth kinetics of progressive lesions in OPD patients. In this case, it was only calculated from measures of progressive lesions.

### 2.4. Statistical Analyses

Patients were classified into 2 groups: “primary resistance” and “Progressive Disease after Durable Clinical Benefit” (“PD after DCD”). We assessed post-progression survival (PPS) and sites of progression. An exploratory analysis was performed using TGR to assess global growth kinetics of cancer and remaining benefit of immunotherapy beyond PD after DCB.

The primary objective was to assess the PPS in the “PD after DCB” group and compare it with primary resistance on anti-PD-1/PD-L1 monotherapy. Secondary objectives were to analyze the patterns and kinetics of PD after DCB in order to define a therapeutic strategy.

The PPS endpoint was defined as the time elapsed between date of progression on ICI and date of death or last contact. Survival rates were estimated using the Kaplan–Meier method, and statistical significance of PPS differences was used in the Log-rank test. We used a Cox regression full model of PPS, including the following covariables of interest to obtain adjusted hazard ratios (HR) for death with their 95% confidence intervals (95% CI): age at progression, sex, histology, treatment line of ICI, ECOG performance status (0–1 vs. ≥2) at ICI initiation and at progression, and disease stage (localized vs. metastatic).

Characteristics of individuals were compared between the “PD after DCB” and “primary resistance” groups using the Wilcoxon rank sum test for numerical values and the exact Fisher test for categorical values. PD-L1 TPS categories were compared using the weighted global Fischer test. Follow-up completion was assessed using Clarck’s index (Lancet, 2002). Covariable effect sizes in multivariable model were statistically significant if their *p*-values were < 0.05. Statistical analyses were performed using R.4.1.3.

## 3. Results

### 3.1. Patients Characteristics

The charts of 189 aNSCLC patients were reviewed; all had received a first course of PD-1/PD-L monotherapy starting between June 2015 and October 2019 and presented a RECIST 1.1 progression after at least 6 weeks of exposure. Of them, 148 patients were included in this study; reasons for exclusion are detailed in the study diagram (Figure 1).

Progression occurred in 43 of 61 patients with initial DCB and after a median time of 11.4 months (95% CI: 9.4–16.4 months).

Included patients were split into “primary resistance” (*n* = 105) and “PD after DCB” (*n* = 43) groups. This last group was reclassified according to the SITC criteria as “late progression” if patients experienced PD after a break from therapy (*n* = 10/43; 23.3%), otherwise as “secondary resistance” (*n* = 33/43; 76.7%).

The baseline characteristics of the included patients are shown in Table 1. As of PD, the global median follow-up index was, respectively, 0.99 (95% CI: 0.44–0.99) and 0.99 (95% CI: 0.98–0.99) in “PD after DCB” and “primary resistance” groups. PD-L1 score categories were balanced between the two groups. Out of 15 patients with a known baseline TPS ≥ 50%, 9 (60%) received anti-PD-1 as their first line of treatment.

### 3.2. Comparative Analysis of PPS in “PD after DCB” and “Primary Resistance” Groups

Patients with initial DCB followed by resistance to anti-PD-1/PD-L1 had a longer PPS than patients with primary resistance as follows: median PPS of 21.3 months (95% CI: 18.5–36.3) and 5.2 months (95% CI: 2.6–6.5) (*p* < 0.0001), respectively (Figure 2). The adjusted HR for death in “PD after DCB” vs. primary resistance was 0.14 (95% CI: 0.07–0.30) (Figure 3).

### 3.3. OPD Prevalence in PD after DCB and Association with PPS

The analysis of the “PD after DCB” group showed that 33 of the 43 patients (76.7%) presented a progression in ≤3 lesions and were thereby classified as OPD. Conversely, 10/43 (23.3%) patients had >3 progressive lesions and were classified as non-OPD. Strikingly, most OPD patients (58%) displayed progression in only one lesion. Median PPS was 22.0 months (95% CI: 20.5-NR) and 19.2 months (95% CI, 12.2-NR), respectively, in OPD and non-OPD, with univariable HR for death in OPD = 0.65 (95% CI: 0.27–1.57)**.** OPD was found to be a protective factor upon multivariable analysis within the “PD after DCB” group, adjusting for PD-L1 expression and PFS under anti-PD-1/PD-L1 (significant factors in the full model), with a lower hazard for death after DCB compared to non-OPD: adjusted HR for death = 0.37 (95% CI: 0.09–1.59).

### 3.4. TGR Evaluation in the “PD after DCB” Group

We first performed an exploratory multivariable analysis to assess the predictive potential of PPS of the TGR measured at PD for evaluable patients in the “PD after DCB” group (*n* = 30). Although yielding a wide confidence interval due to the small sample size, this analysis showed a substantially decreased PPS for patients having a TGR at PD > 25%/month after adjusting for age, ECOG at PD, and tumor histology: adjusted HR for death = 2.5 (95% CI: 0.6–10.3).

Another exploratory comparison of the TGR was also conducted in patients of the “PD after DCB” group when the required imaging data were available at baseline and at progression. In at least 19/43 (44.2%) patients, the TGR assessed at PD was either null/negative or inferior to the TGR measured at baseline before ICI (Appendix A). These TGR results suggest an ongoing benefit of the anti-PD-1/PD-L1 treatment on target lesions despite PD. Overall, the TGR analysis in the “PD after DCB” group showed a low kinetic progression with a median volume growth of 2.1%/month and an interquartile range (IQR) of 19.6%/month. In OPD assessable patients (*n* = 24), the median TGR measured specifically on progressive lesions was 24.6%/month (IQR 17.8%/month) (Appendix A), which is to say that progressive lesions in OPD patients had a median volume growth rate of about one-quarter per month.

### 3.5. Sites of Progression in the “PD after DCB” Group

The main sites of PD for the “PD after DCB” group were lungs for 19/43 (44.2%), lymph nodes for 15/43 (34.9%), and central nervous system (CNS) for 5/43 (11.6%) patients. PD occurred exclusively in lymph nodes in 11/43 (25.6%) patients, consistent with previously published observations [23]. At least 8 of these 11 cases showed no extra lymphatic progression at the next imaging. There were no substantial differences regarding these patterns of progression between secondary resistance and late progression as defined in the SITC. In 31 out of 43 (72.1%) patients, progression occurred exclusively in pre-existing lesions (i.e., lesions that had been already identified prior to ICI treatment). Interestingly, spreading to previously non-affected organs occurred more frequently in primary resistance than in the “PD after DCB” group (Table 2).

### 3.6. Disease Spreading after OPD

When OPD was diagnosed after DCB, the next imaging showed no new lesion in 85.7% of cases (24/28), with a median time between PD imaging and the next imaging of 2.38 months (Appendix A).

Patients were more often eligible to receive subsequent anti-neoplastic treatments in the case of PD after DCB than with primary resistance (Table 2). Subsequent treatments are detailed in Appendix A.

## 4. Discussion

Our study shows that progression is frequent (70.5% of cases) after DCB in aNSLC patients treated with anti-PD-1/PD-L1 monotherapy and that this event occurs after a median of 11.4 months (95% CI: 9.4–16.4 months). The comparison between the two groups “primary resistance” and “PD after DCB” demonstrates diverging outcomes and patterns of progression. This is, to our knowledge, the first study to provide a direct comparative assessment and to use the SITC criteria. This is also the first study to explore the feasibility and usefulness of TGR in the setting of progressive aNSCLC after DCB.

### 4.1. Features of Progression

Firstly, we found a drastic difference in PPS depending on whether or not PD occurs after DCB. In addition to extended PPS, most patients of the “PD after DCB” group (36/43 –83.7%) had a performance status ≤ 1 at PD. Along with other studies, we also demonstrated, that PD after DCB is frequently exclusively located in preexisting lesions (72.1%) [23,24]. Moreover, no emergence of other metastatic sites was seen in the short/medium term. This feature has important implications: it suggests that in individuals experiencing DCB, proactive treatment of residual lesions with locoregional strategies might eliminate the potential sources of acquired resistance and prolong indefinitely the benefit of immunotherapy.

OPD appears as the most frequent pattern of “PD after DCB” (76.7%) and as a good prognosis factor in this situation, consistent with other studies [23,24,25,26,27]. OPD is theoretically defined as a progressive disease with the growth of a restricted number of lesions. It is distinct from the oligometastatic status, although some overlap exists between both entities [19,28].

Therefore, it seems to be of major importance to distinguish the OPD and non-OPD categories among patients experiencing PD, as illustrated in the individual cases in Figure 4 [24]. However, there is yet no consensus about a precise number of progressive lesions that should be used to categorize OPD or non-OPD patients. It is largely admitted that such a number should be based on the possibilities of local treatment [29]. For this reason, the limit of ≤3 progressive lesions were chosen for our study, which is widely used in aNSCLC publications and reflects the practice in our center [24,25,27]. This is also consistent with the milestone randomized phase II trial of local consolidative therapy of oligometastatic NSCLC from Gomez et al., which enrolled patients with a maximum of three lesions [30]. Of note, recent studies have used higher thresholds [31]. To date, our study is the first to show that most OPD occurring after DCB (85.7%) experienced no further spreading of aNSCLC in the short/medium term. These important data strengthen the interest in local treatment.

### 4.2. Treatment beyond Progression

#### 4.2.1. Systemic Treatment

Our study brings new data to support the continuation of ICI beyond progression via the assessment of TGR. Indeed, a persistent effect on tumor growth, in spite of PD, is suggested when the TGR at PD, calculated with the help of additional imaging performed a couple of weeks later on ICI, is still inferior to the TGR at the time of ICI initiation. Globally, TGR analysis in the “PD after DCB” group showed a low kinetic progression, with a median volume growth of 2.1%/month. This suggested, in some patients, a persistent benefit of immunotherapy in target lesions. These results provide useful information to decide on ICI continuation rather than treatment line switch [32]. This is, to our knowledge, the first published study to apply the TGR in the setting of progressive aNSCLC after DCB, although it has been used in other studies of NSCLC patients treated with ICI [17,18,33,34]. The TGR appears as a promising and feasible tool, given the current paucity of the objective criteria to establish a persistent benefit of ICI after PD.

In addition to its role in measuring the residual benefit of PD-1/PD-L1 blockade, the TGR could also be investigated as a predictive factor for PPS in the setting of PD after DCB in future studies in the same way as what has been previously performed at the baseline of immunotherapy in NSCLC [17,18].

It should be noted that the TGR relies only on RECIST measures of target lesions. For this reason, a negative TGR can be observed although there is a tumor progression in non-target lesions. Nonetheless, this would still indicate a persistent benefit, at least on the proportion of disease burden residing in target lesions.

The continuation of immunotherapy in OPD patients should also be considered for several reasons. The discontinuation of PD-1/PD-L1 blockade in the context of OPD could lead to the subsequent progression of currently non-progressive lesions in the absence of an autonomous efficient anti-tumor immune response, especially in patients who presented with multiple other disease sites before immunotherapy treatment. At the preclinical level, studies suggest that the antitumor immune response harnessed via anti-PD-1/PD-L1 monotherapy relies on pre-existing CD8+ exhausted lymphocytes requiring a sustained blockade of their inhibitory receptors to remain functional, as opposed to a de novo immune response that would be independently self-maintained [35]. At the clinical level, our 10 cases of late progression after PD-1/PD-L1 blockade discontinuation (for toxicity or practical reasons) also support this theory. On the other hand, OPD patients achieving complete clearance of clinically detectable disease could be managed differently, as suggested in observational studies of immunotherapy discontinuation [36].

#### 4.2.2. Local Treatment

In the particular setting of oligoprogressive disease, using local treatments (surgery or stereotactic body radiation) of locally progressive lesions and continuing the ongoing ICI is an increasingly accepted strategy. This strategy is known to be beneficial in oncogene-addicted aNSCLC, but prospective studies are needed in the case of ICI treatment [37]. In our study, two-thirds of patients presented an oligoprogressive pattern, and TGR appeared of major interest when specifically applied to OPD lesions. For instance, a given TGR of OPD lesions would prompt quick initiation of SBRT before the gross tumor volume exceeds the technical possibilities. So, the estimation of TGR for OPD lesions provided in this study may also be useful in deciding the time and technic for such a treatment.

In spite of these theoretical advantages, a minority of PD after DCB (4/43–9.3%) was treated with local therapies in our cohort (Appendix A). This might be due, to some extent, to the lack of follow-up data available at that time for clinicians to support these strategies in “PD after DCB” and emphasizes the relevance of our longitudinal study of this subset of patients. This specific situation is also currently not addressed in international guidelines, which provide the generic recommendation of switching immunotherapy for systemic chemotherapy in the case of progression [38,39]. Our findings that most patients with OPD after DCB did not present with additional lesions on the medium-term follow-up and had extended PPS are reassuring. We hope that such data, along with technical advances in local therapies, will encourage their use in the setting of OPD after DCB.

### 4.3. Considerations for the Mechanisms of Resistance

In the vast majority of patients treated with anti-PD-1/PD-L1, the control of the disease is temporary. The mechanisms of resistance to anti-PD-1/PD-L1 still need to be elucidated, although multiple hypotheses based on preclinical models have been formulated [40]. Resistance after DCB or secondary resistance could rely on distinct but also common mechanisms to primary resistance [41]. In this context, the implementation of clinical, pathological, and biological data gathered in cohorts of patients treated with anti-PD-1/PD-L1 monotherapy seems pivotal to selecting and delving into the most relevant of these hypotheses.

#### 4.3.1. Sites of Progression

The existence of preferential sites of progression after DCB could indicate specificities of the microenvironment favoring a delayed immune escape. In the “PD after DCB” group, the most frequent progressive sites were the lungs, lymph nodes, and the CNS. The same hierarchy was observed in secondary resistance and late progression subgroups, as defined in the SITC. A similar repartition has also been reported by Heo et al. in a cohort of aNSCLC mostly treated with anti-PD-1/PD-L1 monotherapy and using a fairly close definition of secondary resistance [23]. It has been hypothesized on the basis of preclinical models that the lung microenvironment could be prone to secondary resistance [42,43,44]. The clinical data presented here are consistent with this hypothesis. Interestingly, disease spreading to new organs was more frequent in primary resistance than in PD after DCB (Table 2).

#### 4.3.2. Lymphatic Progression after Durable Clinical Benefit Is a Distinct Entity

As previously reported, a substantial proportion (25.6%) of PD after DCB occurred only in lymph nodes [23]. It could be objected that these lesions simply witness a subclinical growth of the disease in neighboring organs. Nevertheless, the observation in our study that the majority of these PD showed no involvement of other organs at the next imaging supports the theory that secondary resistance limited to lymph nodes is a distinct biological entity. It has been hypothesized that this could be due to the long-term effects of PD-1/PD-L1 blockade on intra-lymphatic antitumor immune cells, either on their survival or on chemokine signals enabling their location in nodes [25,27].

#### 4.3.3. PD after DCB Is a Localized Phenomenon Arising in Preexisting Lesions

As reported in previous studies, we found that the majority of cases of PD after DCB occurred as OPD in pre-existing lesions [23,24,26,45]. Here again, the biological relevance of our observations is strengthened via the careful review of imaging performed after the first PD assessment. This suggests the sustainability of the systemic immune surveillance under anti-PD-1/PD-L1 monotherapy, along with the localized evolution of a few pre-existing lesions toward immune evasion or immune suppression. This localized resistance could arise, for instance, from cancer cells immunoediting or from the development of immunosuppressive conditions within a given lesion [41]. Furthermore, PD after DCB is less prone to arise in new organs compared to primary resistance. This might indicate an evolution of aNSCLC under long-term PD-1/PD-L1 blockade toward better fitness in already invaded tissue rather than the colonization of completely new tumor microenvironments.

### 4.4. Other Considerations about Strengths and Limitations of This Study

In contrast with other publications using non-consensual and heterogeneous criteria, we used the definitions provided by the SITC in order to contribute to the necessary harmonization of clinical and translational research about immunotherapy resistance. This makes possible the collection of homogeneous data in the future and aggregated and reproducible analyses. Also, for reproducibility reasons, we had a strict definition of PD based on RECIST1.1 that might theoretically encompass patients undergoing pseudoprogression instead of true biological disease progression. However, pseudoprogression is a rare phenomenon in NSCLC, and we observed no instances of such a situation in our sample of patients [46].

The “PD after DCB” group comprised two subgroups of secondary resistance (*n* = 33) and late progression (*n* = 10), which are worth differentiating. Indeed, aNSCLC progression in the setting of late progression could rely on different mechanisms from secondary resistance, given that it would happen off therapy. For instance, late progression could be due to impaired global immune surveillance after PD-1/PD-L1 blockade discontinuation, indicating that the dysfunction of antitumor immune cells is not definitively reversed with anti-PD-1/PD-L1. Interestingly, in our series, although based on small numbers, there were no striking differences in the patterns of progression between these two subgroups.

Nevertheless, the SITC criteria might be criticized, concerning the 6-month threshold distinguishing secondary from primary resistance. As explained by SITC experts, this minimal period of DCB is necessary to exclude aNSCLC growing slowly under ICI treatment without any prior benefit but classified as stable disease (SD) per RECIST1 [11]. But, such a threshold may misclassify some aNSCLCs as being primary resistant, although they have shown prior response to immunotherapy before PD and might be relevantly categorized as being secondary resistant. Schoenfeld and colleagues have recently challenged these definitions and demand their adaptation [47]. This controversy also explains the growing interest in TGR in order to improve the definition of secondary resistance and better identify patients presenting the benefit of ICI prior to resistance. The TGR has been used by Ferté et al. to evaluate the benefit of anticancer drugs in early-phase clinical trial data [48]. However, due to the retrospective design of our study, the TGR could not be collected for all patients, which prevented us from including this parameter in all multivariable analyses.

Some previous studies pooled data about anti-PD-1/PD-L1 monotherapy and combinations, including anti-CTLA-4, chemotherapy, or tyrosine kinase inhibitors. For instance, Schoenfeld et al. analyzed a cohort of patients treated either with anti-PD-1/PD-L1 monotherapy or combined with anti-CTLA-4, assuming that the patterns and mechanisms of resistance should be similar between both types of therapies but without providing a subgroup analysis and in contradiction to what is known from the respective mechanisms of action of these drugs [24,35]. Our series provides more homogeneous data, as we only included patients under anti-PD-1/PD-L1 monotherapy, thus increasing the relevance of biological hypotheses in secondary resistance. In addition, our cohort consists mostly of Caucasian patients with data that might biologically better apply to Western countries’ populations. For instance, one of the largest cohorts of secondary resistance in aNSCLC (*n* = 51 patients under anti-PD-1/PD-L1 monotherapy) originates from Korea and comprises a high proportion of EGFR-mutated NSCLC (≥17.4%) [23]. On the other hand, some heterogeneity in our cohort could arise from the inclusion of both aNSCLC receiving anti-PD-1/PD-L1 monotherapy as a first line of treatment (in case of ≥50% PD-L1 expression) and already pre-treated patients. This could also be a potential confounder with regard to the association between the type of resistance and PPS. Indeed, trials such as the KEYNOTE-024 suggest better outcomes of frontline immunotherapy rather than as a second (or later) line, though it is not clear if this factor also influences the PPS in addition to extending the PFS on immunotherapy [5]. More generally, the number of previous lines of treatment is a classical prognosis factor in aNSCLC. The size of our sample did not allow us to efficiently compare both types of resistance within each stratum of the ICI line. For these reasons, we incorporated this variable in the Cox multivariable PPS model (Figure 3), which still resulted in a strong association between PPS and the type of resistance while adjusting for the immunotherapy line.

It can also be noted that our imaging review of PD after DCB was performed by the same two trained investigators on consecutive imaging, which improves the confidence in the patterns of progression demonstrated here. Otherwise, these patterns could have been confounded by the inter-observer variability or the insufficient sensitivity of morphological imaging at a unique time point.

A substantial proportion of patients (42.6%) had no PD-L1 TPS evaluation prior to ICI. This is explained by the fact that a large share of our cohort received Nivolumab as a ≥2nd-line treatment, which was indicated regardless of TPS, before first-line Pembrolizumab approval. Therefore, PD-L1 scoring was not routine practice at that time. To our knowledge, all other publications on this topic face the same issue, having many TPS values missing or not providing them at all [23,24,26,27,45]. However, some exploratory pathological analyses of PD-L1 performed in our center enabled us to provide TPS values even for some of the ≥2nd-line Nivolumab-treated patients.

## 5. Conclusions

We provide here the first real-world comparative analysis of the types of resistance to immunotherapy in aNSCLC.

“PD after DCB” under anti-PD-1/PD-L1 monotherapy is shown to be an independent factor of extended PPS and displays distinct patterns. The distinctive feature of “PD after DCB” must be taken into account when discussing therapeutic strategies in the progression of aNSCLC patients. As previously reported, OPD status tends to be associated with better prognosis among PD after DCB patients. Our longitudinal analysis brings objective data to support locoregional treatment in patients experiencing DCB before or after subsequent progression. TGR is a promising tool to guide decisions regarding ICI continuation and local treatments and to refine the secondary resistance definition. Altogether, these data shall contribute to a better biological understanding and clinical management of aNSCLC secondary resistance to immunotherapy (Figure 5). It will profitably add up with other studies and improve significance with larger sample sizes.

## Figures and Tables

**Figure 1 cancers-15-05587-f001:**
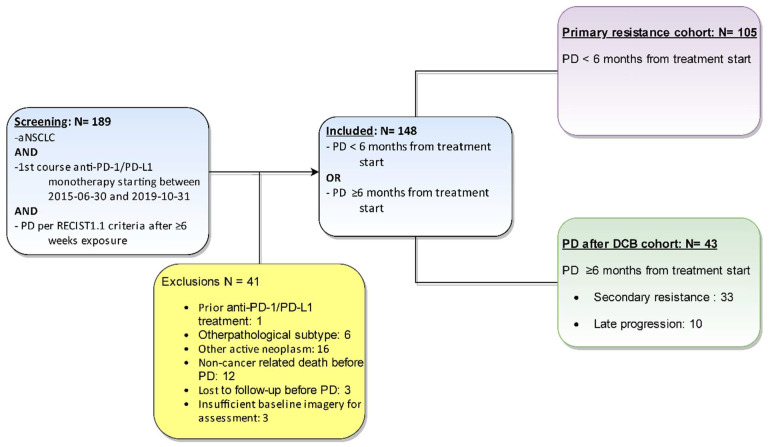
Study flowchart. aNSCLC: advanced non-small cell lung cancer; PD-1: programmed cell death protein 1; PDL-1: programmed death ligand 1; PD: progressive disease; RECIST: Response Evaluation Criteria in Solid Tumors.

**Figure 2 cancers-15-05587-f002:**
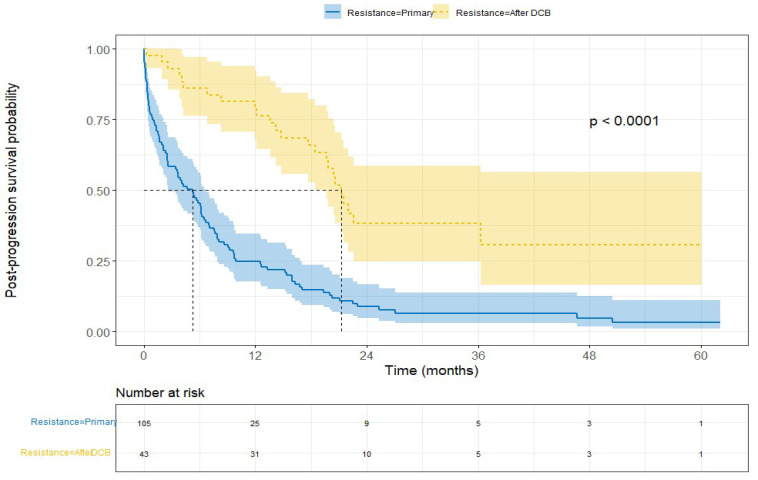
Comparison of post-progression survival in “PD after DCB” versus “primary Resistance”. Kaplan–Meier survival curves showing post-progression survival of “PD after DCB” versus “primary Resistance” groups. *p*-value for Log-rank test.

**Figure 3 cancers-15-05587-f003:**
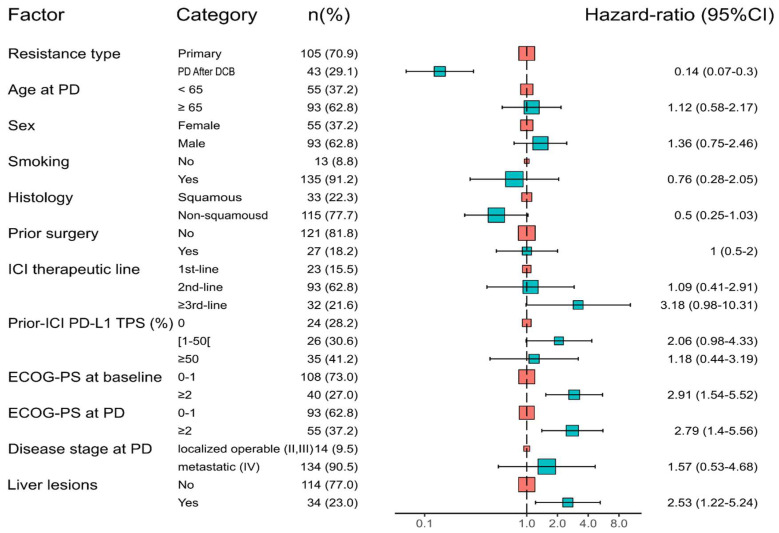
Cox multivariable model for post-progression survival (PPS) in “PD after DCB” versus “primary Resistance”. Forest plot showing HR for death in “PD after DCB” versus “primary Resistance” and other significant covariables. PD: progressive disease; ICI: immune checkpoint inhibitor; PS: performance status.

**Figure 4 cancers-15-05587-f004:**
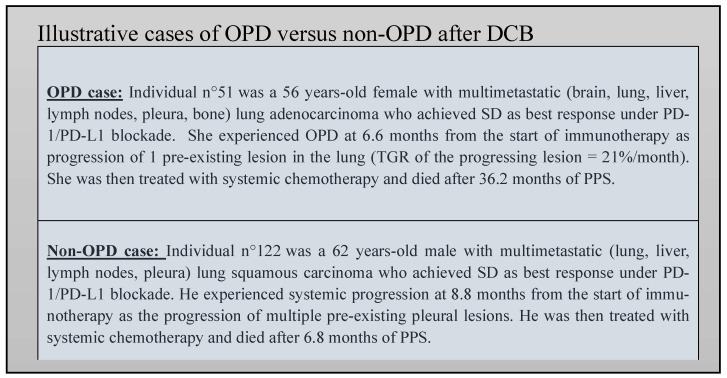
Illustrative cases of OPD versus non-OPD after DCB.

**Figure 5 cancers-15-05587-f005:**
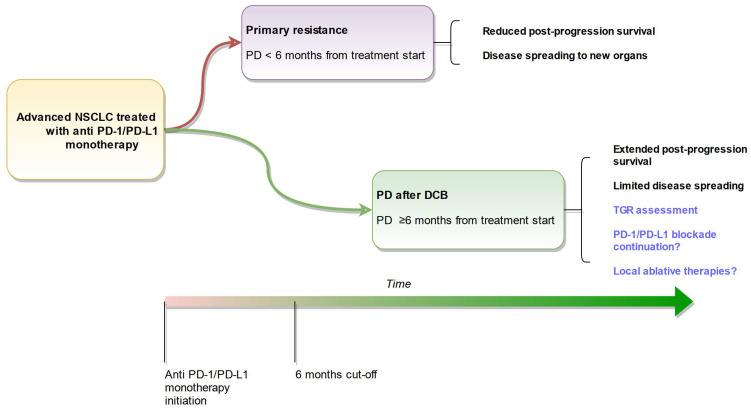
Summary of findings and perspectives.

**Table 1 cancers-15-05587-t001:** Characteristics of study population.

	Primary Resistance (*n* = 105)	PD After DCB (*n* = 43)	Overall (*n* = 148)	*p*-Value
Age at PD (years)				
Median (IQR)	70 (±14)	67 (±12)	69 (±14)	0.999
Sex (*n* (%))				
Female	39 (37%)	16 (37%)	55 (37%)	1
Male	66 (63%)	27 (63%)	93 (63%)	
Smoking (*n* (%))	95 (90%)	40 (93%)	135 (91%)	
Histology (*n* (%))				
Squamous	25 (24%)	8 (19%)	33 (22%)	0.827
Non-squamous	80 (76%)	35 (81%)	115 (78%)	
Prior surgery (*n* (%))				
Yes	18	9	27	0.864
No	87	34	121	
Disease stage at PD (*n* (%))				
Localized non resectable (II, III)	8 (8%)	6 (14%)	14 (9%)	0.465
Metastatic (IV)	97 (92%)	37 (86%)	134 (91%)	
ICI therapeutic line (*n* (%))				
1st line	14 (13%)	9 (21%)	23 (16%)	0.778
2nd line	69 (66%)	24 (56%)	93 (63%)	
≥3rd line	22 (21%)	10 (23%)	32 (22%)	
Immune checkpoint inhibitor (ICI) (*n* (%))				
Atezolizumab	5 (5%)	1 (2%)	6 (4%)	0.864
Nivolumab	81 (77%)	31 (72%)	112 (76%)	
Pembrolizumab	19 (18%)	11 (26%)	30 (20%)	
Prior ICI TPS value (%) (*n* (%))				
0	18 (17%)	6 (14%)	24 (16%)	0.718
[1–50]	18 (17%)	8 (19%)	26 (18%)	
≥50	20 (19%)	15 (35%)	35 (24%)	
Missing	49 (46.7%)	14 (32.6%)	63 (42.6%)	
Disease sites at ICI start (*n* (%))				
Adrenal	30 (29%)	7 (16%)	37 (25%)	
Pleura	39 (37%)	20 (47%)	59 (40%)	
Liver	26 (25%)	8 (19%)	34 (23%)	
Bone	52 (50%)	15 (35%)	67 (45%)	
Lymph nodes	86 (82%)	36 (84%)	122 (82%)	
Other	12 (11%)	2 (5%)	14 (9%)	
ECOG-PS at ICI start (*n* (%))				
0–1	76 (72%)	32 (74%)	108 (73%)	0.984
≥2	29 (28%)	11 (26%)	40 (27%)	
ECOG-PS at PD (*n* (%))				
0–1	57 (54%)	36 (84%)	93 (63%)	0.003
≥2	48 (46%)	7 (16%)	55 (37%)	

At ICI start: at initiation of the anti-PD-1/PD-L1 treatment; At PD: at progression after anti-PD-1/PD-L1 initiation; ICI: immune checkpoint inhibitor; TPS: tumor proportion score; PS: performance status; PD: progressive disease; DCB: durable clinical benefit.

**Table 2 cancers-15-05587-t002:** Summary of patients’ outcomes stratified by resistance group.

	Primary Resistance (*n* = 105)	PD after DCB (*n* = 43)
Disease spreading at new sites (*n*(%))		
Liver	8 (7.6)	1 (2.3)
Nodes	4 (3.8)	1 (2.3)
Pleura	10 (9.5)	0 (0)
Adrenal	5 (4.8)	1 (2.3)
Bones	6 (5.7)	0 (0)
Brain	7 (6.7)	1 (2.3)
Any cancer treatment after PD (*n*(%))		
	67 (63.8)	37 (86.0)
Median PPS (months (95%CI))		
	5.2 (2.6–6.5)	21.3 (18.5–36.3)

PD: progressive disease; DCB: durable clinical benefit.

## Data Availability

The data presented in this study are available on request from the corresponding author. The data are not publicly available due to the local clinical data sharing policy.

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
