# Peer review of "Longitudinal Study of Advanced Non-Small Cell Lung Cancer with Initial Durable Clinical Benefit to Immunotherapy: Strategies for Anti-PD-1/PD-L1 Continuation beyond Progression"

_cancers, 2023, doi:10.3390/cancers15235587_

Round 1
Reviewer 1 Report
Comments and Suggestions for Authors
CONGRATS,NICE WORK!
ABSTRACT : OK
INTRODUCTION : MORE DATA IN EARLY vs ADVANCED STAGE.
MATERIALS :OK
RESULTS :OK
DISCUSSION : OK
A PARAGRAPH TO EXPLAIN THE PD-L1 PATHOLOGY COUNT (CLONE,TPS/CPS,% etc)COULD GIVE A POINT IN FAVOR!
Author Response
- Introduction : More data in early versus advanced stage
The following elements have been added in the revised version of the manuscript:
Localized NSCLC is a curable disease albeit a remaining risk of localized and/or distant recurrence after treatment, increasing with stage (American Cancer Society 2019). Advanced stages of the disease (aNSCLC) are associated with poor prognosis in historical cohorts with a median survival around 3 months in untreated patients and 8 months under conventional chemotherapies.
- PD-L1 pathology:
The following elements have been added in the revised version of the manuscript (“Methods, 2. Clinical, pathological and radiological data”):
PD-L1 expression prior anti-PD-1/PD-L1 treatment was quantified by a specialized pathologist using PD-L1 immunohistochemical staining of tumor samples (antibody: PharmDx 22C3, Agilent), and expressed as percentage of positive tumor cells (Tumor Proportion Score - TPS).
Reviewer 2 Report
Comments and Suggestions for Authors
The authors conducted a retrospective study analyzing the factors contributing to durable clinical benefits of immunotherapy. This study is interesting and important for physicians in predicting the effects of immunotherapy in patients with NSCLC. However, there were several flaws in the manuscript.
Firstly, the manuscript is not well-structured. The authors included several definitions in the introduction section that should have been placed in the Methods Section. Ethical considerations are typically presented at the beginning of the Methods section.
Secondly, several variable factors that emerged during the treatment, such as TGR, resistance type, ICI therapeutic line, ECOS-PS at PD, disease stage at PD, and others, were used to predict the hazard ratio. Including these variables in the analysis made the results unreliable.
There were also several minor mistakes in the figures and table illustrations.
Author Response
- Firstly:
Thank you for this remark, ethical considerations have been moved to the beginning of the Methods section.
- Secondly:
We apologize if there is some misunderstanding in the design of the study. We tried to clarify it in the new version of the manuscript. Here is a short explanation:
The aim of the study was not to observe the individual trajectories of every NSCLC patient treated with ICI, from the start of their ICI treatment. Instead, our study focused on patients that had resistance to ICI, and their trajectories from the moment this resistance was documented. The primary goal was to evaluate the association between the type of progression (primary resistance vs PD after DCB) and post-progression survival (PPS); for this, we also took into account other clinical and biological factors present at the moment of this first progression and possibly related to PPS. Therefore, patients were followed from their 1st progression after a 1st course of ICI, up to death or date of last contact, that is to say the post progression period. Thus, the factors present at the 1st progression that you mentioned, including the type of resistance to ICI, are baseline co-variables for this PPS period, but not longitudinal (they were not measured during the follow-up period of our study).
Indeed, in “Table 1. Patients characteristics.”, some characteristics have been confusingly called “Disease sites at baseline” and “ECOG-PS at baseline”. In these cases, the term “at baseline” was inappropriate because our baseline corresponds to the 1st progression under ICI treatment (called “PD” in Table 1). Hence, the inappropriate “at baseline” term has been replaced by “at ICI start” in the revised version of the manuscript.
Reviewer 3 Report
Comments and Suggestions for Authors
The manuscript entitled ‘Longitudinal Study of Advanced Non-small Cell Lung Cancer With Initial Durable Clinical Benefit to Immunotherapy: Strategies for Anti-pd-1/Pdl-1 Continuation Beyond Progression’ describes the difference between initial progressive disease (PD) and PD after durable clinical benefit (DCB), comparing the post-progression survival (PPS) and the patterns of progression.
The study was well-organized and interesting in that the authors tried to elucidate the patients who would benefit from beyond PD use of immune checkpoint inhibitors (ICIs).
However, there are some concerns to be solved.
Major:
1) This study evaluated any line of ICI treatment, including the first-line ICI monotherapy irrespective of PD-L1 expression status. As shown in KEYNOTE-024 trial, those patients with NSCLC with PD-L1 expression of ≥50% who received platinum-based chemotherapy in the first-line setting showed shorter progression-free survival (PFS) and overall survival (OS) compared to the pembrolizumab monotherapy. Therefore, first-line pembrolizumab or atezolizumab treatment is different from the second-line or later treatment for NSCLC with PD-L1 expression of ≥50%. Then, analyzing the first-line ICI monotherapy with the second-line or later would lead to a strong bias in this study, affecting PFS, OS, and PPS.
Thus, the author should pay attention to this point, for example, by excluding the first-line ICI monotherapy in this study.
2) The tumor growth rate (TGR) was used to evaluate the tumor kinetics in this study. Because TGR is not familiar to the potential readers, the author should describe its definition in detail in the Materials and Methods section, not leaving the explanation to Supplementary Figure S1.
3) Considering the definition of TGR, the authors should compare the difference between pseudo-PD from true-PD. I think the authors should address this issue in the Discussion section.
4) Discussion section: the contents are too redundant, and the contents should be described more concisely.
Minor:
1) Title: ‘Anti-pd-1/Pdl-1’ is not common, and I think ‘Anti-PD-1/PD-L1’ would be better.
2) Line 149: ‘PD after PD’ is a typographical error.
3) The hyphen in ‘PD-1/PD-L1’ are lacking. Therefore, they should be corrected.
4) The term ‘tumor growth rate’ appears after abbreviating this term as TGR.
Comments on the Quality of English LanguageThere are some grammatical errors.
Author Response
We would like to thank you for your careful reading of our manuscript and valuable remarks that helped improving its quality. Below is a point-by-point response.
Major:
1) This study evaluated any line of ICI treatment, including the first-line ICI monotherapy irrespective of PD-L1 expression status. As shown in KEYNOTE-024 trial, those patients with NSCLC with PD-L1 expression of ≥50% who received platinum-based chemotherapy in the first-line setting showed shorter progression-free survival (PFS) and overall survival (OS) compared to the pembrolizumab monotherapy. Therefore, first-line pembrolizumab or atezolizumab treatment is different from the second-line or later treatment for NSCLC with PD-L1 expression of ≥50%. Then, analyzing the first-line ICI monotherapy with the second-line or later would lead to a strong bias in this study, affecting PFS, OS, and PPS.
Thus, the author should pay attention to this point, for example, by excluding the first-line ICI monotherapy in this study.
We totally agree on the point that patient outcomes can vary strongly depending on their current treatment lines and that it can be a source of bias when pooling together patients from different lines, also in the context of immunotherapy. We also understand from your remark that you would expect patients to have a greater efficacy of anti-PD1 when it’s given as a first-line, as opposed as later lines. However, we might have not been clear enough:
- Indeed, pivotal trials suggest that patients achieve better results on immunotherapy when it’s given from the first-line, rather than from the ≥2nd line (although in these trials not every eligible patient of the control chemotherapy arm had received Nivolumab upon progression as they should have, in order to definitely establish that frontline immunotherapy is better than sequential treatment regarding PFS2 or OS). We focus on outcomes in the particular period of post-progression, and do not compare the efficacy and survival on immunotherapy before this point.
- We tried to adjust for the potential influence of the ICI treatment line on the subsequent PPS by incorporating it in the Cox multivariable PPS model (the “ICI therapeutic line” covariable in Fig.3). Of course, this method has known limitations, but has nonetheless proven to be useful in the context of observational studies. Hence, our analysis suggests that the PPS of primary resistance still differs significantly from PD after DCB, after adjusting for the immunotherapy line. Unfortunately, the size of our sample did not allow to efficiently compare both types of resistance within each stratum of ICI line.
2) The tumor growth rate (TGR) was used to evaluate the tumor kinetics in this study. Because TGR is not familiar to the potential readers, the author should describe its definition in detail in the Materials and Methods section, not leaving the explanation to Supplementary Figure S1.
Thanks you for this remark, we give more detail about the TGR methodology in the revised version of the manuscript (3. Tumor kinetics evaluation by Tumor Growth Rate (TGR) ):
The global growth kinetics of cancer was estimated by the TGR, as previously pub-lished.[17,18] This method allows to estimate the percentage of tumor burden increase per month. For this purpose, the global tumor burden of a given patient is modeled as a virtual tumor sphere growing exponentially over time, with its volume Vt given at time “t” by the formula Vt = V0.exp(TG.t). The diameter of this virtual tumor sphere equals the sum of longest diameters of target lesions by RECIST 1.1. Thus, having the diameter at two different time points, the parameter “TG” can be retrieved from the formula (Supplemen-tary Fig. S1).
3) Considering the definition of TGR, the authors should compare the difference between pseudo-PD from true-PD. I think the authors should address this issue in the Discussion section.
Thank you for this remark, we address this in the revised version of the manuscript (4. Other considerations about strengths and limitations of this study):
Also, for reproducibility reasons we had a strict definition of PD based on RECIST1.1 that might theoretically encompass patients undergoing pseudoprogression, instead of true biological disease progression. However, pseudoprogression is a rare phenomenon in NSCLC and we observed no instances of such situation in our sample of patients.[46]
4) Discussion section: the contents are too redundant, and the contents should be described more concisely.
We strived to be concise in the revised version of the manuscript. However, please be aware that other reviewers required us to develop several points of the discussion, so we hope for your indulgence.
Minor:
1) Title: ‘Anti-pd-1/Pdl-1’ is not common, and I think ‘Anti-PD-1/PD-L1’ would be better.
Thank you, this was a typo.
2) Line 149: ‘PD after PD’ is a typographical error.
corrected.
3) The hyphen in ‘PD-1/PD-L1’ are lacking. Therefore, they should be corrected.
corrected.
4) The term ‘tumor growth rate’ appears after abbreviating this term as TGR.
corrected.
Reviewer 4 Report
Comments and Suggestions for Authors
I wish to express my appreciation for the opportunity to review your scholarly work, which provides valuable insights into a clinically important topic. Nonetheless, I would like to propose some respectful suggestions that aim to enhance the overall quality and impact of this work. These recommendations may contribute to refining your manuscript if you choose to implement them:
Introduction:
· The introduction is thorough. However, the impact of this section could be enhanced by referencing recent clinical practice guidelines that address the management of immunotherapy progression in NSCLC. This would underscore the critical need for empirical evidence to guide clinical recommendations.
· Incorporating additional information regarding the biological underpinnings of oligoprogression could effectively frame it as a distinct phenomenon warranting dedicated analysis.
Methods:
· The methodology section is adequately presented, but some aspects could benefit from more comprehensive detailing. For instance, did you employ blinding for radiologic assessments?
· Concerning survival analyses, a clarification of the statistical censoring approach would add depth to the methodology section.
· Moreover, elucidating the rationale behind the selection of covariates for the multivariable models would facilitate the interpretation of these results. This additional information would strengthen the methodology of the study.
Results:
· The analysis of oligoprogression is a noteworthy highlight. Enhancing this section with case examples may facilitate readers in visualizing the radiologic patterns and their clinical implications.
· To consolidate the findings, the addition of a table summarizing patient outcomes stratified by resistance group could prove valuable.
Discussion:
· The discussion could benefit from a more extensive exploration of the implications of the oligoprogression and tumour growth rate data on immunotherapy strategies beyond the point of progression. This could provide a more comprehensive understanding of the clinical relevance of these findings.
· Exploring the reasons behind the low utilization of local therapies, despite evidence of their benefit, could highlight crucial practice gaps that require further attention and discussion. This would add depth to the analysis of the clinical implications of the study's findings.
In sum, this manuscript is well-crafted and offers valuable evidence to inform the management of NSCLC patients who experience disease progression during immunotherapy. By implementing the above suggestions, you could augment the depth, clarity, and impact of your work, thereby benefiting the readership of the journal.
Author Response
- The introduction is thorough. However, the impact of this section could be enhanced by referencing recent clinical practice guidelines that address the management of immunotherapy progression in NSCLC. This would underscore the critical need for empirical evidence to guide clinical recommendations.
The following elements have been added in the revised version of the manuscript:
Treatment continuation beyond progression is increasingly attempted on the basis of the theoretical pharmacodynamics of immunotherapy. There are currently no clear guidelines about whether, and in which case, PD-1/L1 blockade should be maintained in case of disease progression, although this option was allowed in some of the pivotal trials of first-line immunotherapy and based on the subjective perception of a persistent clinical benefit by investigators (Gandhi et al. 2018; ESMO 2023; NCCN 2023).
- Incorporating additional information regarding the biological underpinnings of oligoprogression could effectively frame it as a distinct phenomenon warranting dedicated analysis.
The following elements have been added in the revised version of the manuscript:
Our limited knowledge of PD after DCB is explained by the scarcity of clinical and bio-logical data and by the use of various definitions of resistance in the literature. Nonetheless, the accumulating observations of particular patterns in the context of secondary resistance, such as oligoprogression (disease progressing in a restricted number of sites), suggest that this entity relies on distinct biological mechanisms and should be thus investigated and managed specifically (see the Discussion section) (Weiss et Sznol 2021).
- The methodology section is adequately presented, but some aspects could benefit from more comprehensive detailing. For instance, did you employ blinding for radiologic assessments?
No blinding was employed for radiological assessment. However, for a given patient, all oncological imaging performed during the follow-up period, and also the one used to classify patients at baseline between primary resistance and PD after DCB, were carefully re-analyzed by the same trained investigator with objective criteria.
- Concerning survival analyses, a clarification of the statistical censoring approach would add depth to the methodology section.
The following elements have been added in the revised version of the manuscript:
For survival analyzes, individuals were right-censored at the date of last contact and if no event had occurred before the date of last survival data collection (20 March 2022).
- Moreover, elucidating the rationale behind the selection of covariates for the multivariable models would facilitate the interpretation of these results. This additional information would strengthen the methodology of the study.
The following elements have been added in the revised version of the manuscript:
We used a Cox regression full model including the following additional covariables to obtain adjusted hazard ratios (HR) with their 95% confidence intervals (95% CI): age at progression, sex, histology, treatment line of ICI, ECOG-Performance status (0-1 vs > 2) at ICI initiation and at progression time, and disease stage (localized vs. advanced/metastatic). These covariables are key influential covariables for survival of aNSCLC reported in the literature.
- The analysis of oligoprogression is a noteworthy highlight. Enhancing this section with case examples may facilitate readers in visualizing the radiologic patterns and their clinical implications.
The Figure 3 have been added in the revised version of the manuscript, Discussion section, 1. Features of progression.
Figure 3: Illustrative cases of OPD versus non-OPD after DCB.
- To consolidate the findings, the addition of a table summarizing patient outcomes stratified by resistance group could prove valuable.
The Table 2 has been augmented in the revised version of the manuscript with a summary of patient outcomes stratified by resistance group.
- The discussion could benefit from a more extensive exploration of the implications of the oligoprogression and tumour growth rate data on immunotherapy strategies beyond the point of progression. This could provide a more comprehensive understanding of the clinical relevance of these findings.
The following paragraph (“2. Treatment beyond progression, 2.1. Systemic treatment”) has been augmented in the discussion:
Our study brings new data to support the continuation of ICI beyond progression through the assessment of TGR. Indeed, a persistent effect on tumor growth, in spite of PD, is suggested when the TGR, calculated with help of an additional imaging performed a couple of weeks later on ICI, is still inferior to the baseline TGR. Globally, TGR analysis in the “PD after DCB” group showed a low kinetic of progression with a median volume growth of 2.1%/month demonstrating a persistent benefit of immunotherapy. These results provide useful information to decide ICI continuation rather than treatment line switch. This is to our knowledge the first published study to apply the TGR in the setting of progressive aNSCLC after DCB, although it has been used in other settings including NSCLC patients treated by ICI. TGR appears as a promising and feasible tool, given the current paucity of objective criteria to establish a persistent benefit of ICI after PD.
In addition to its role as a tool for measuring residual benefit of PD-1/PD-L1 blockade, the TGR could also be investigated as a predictive factor for PPS in the setting of PD after DCB in future studies, in the same way as what has been previously performed at baseline of immunotherapy and more specifically in NSCLC (Champiat et al. 2017; Berge et al. 2019).
It should be noted that TGR relies only on RECIST measures of target lesions. For this reason, a negative TGR can be observed despite tumor progression in non-target lesions. Nonetheless, this would still indicate a persistent benefit in target lesions.
Continuation of immunotherapy in OPD patients should also be considered for several reasons. Discontinuation of PD-1/PD-L1 blockade in a context of OPD could lead to the subsequent progression of the other lesions in the absence of an autonomous anti-tumor immune, especially in patients who presented with multiple other disease sites before immunotherapy treatment. At the preclinical level, studies suggest that the antitumor immune response harnessed by anti-PD-1/PD-L1 monotherapy relies on pre-existing CD8+ exhausted lymphocytes that would require a sustained blockade of their inhibitory receptors to remain functional, as opposed to a de novo immune response that would be independently self-maintained (Wei et al. 2019). At the clinical level, our 10 cases of late progression after PD-1/PD-L1 blockade discontinuation (for toxicity or practical reasons) also support this theory. On the other hand, OPD patients achieving a complete clearance of clinically detectable disease could be managed differently, as observational studies of immunotherapy discontinuation suggest (Ardin et al. 2023).
- Exploring the reasons behind the low utilization of local therapies, despite evidence of their benefit, could highlight crucial practice gaps that require further attention and discussion. This would add depth to the analysis of the clinical implications of the study's findings.
The following paragraph (“2.2. Local treatment”) has been augmented in the discussion:
In spite of these theoretical advantages, a minority of PD after DCB (4/43 – 9.3%) was treated with local therapies in our cohort (Supplementary Table S3). This might be due, to some extent, to the lack of data available at this time for clinician to support local treatments in “PD after DCB”, and emphasizing the relevance of our longitudinal study of this subset of patients. This specific situation is also currently not addressed in international guidelines, providing the generic recommendation of switching immunotherapy for systemic chemotherapy in case of progression (ESMO 2023; NCCN 2023). Our findings that most patients with OPD after DCB did not present with additional lesions on the medium-term follow-up and had extended PPS is reassuring. We hope that such data, along with technical advances in local therapies, will encourage their use in the setting of OPD after DCB.
Round 2
Reviewer 2 Report
Comments and Suggestions for Authors
I have no additional comment on this manuscript
Author Response
.
Reviewer 3 Report
Comments and Suggestions for Authors
I think that the manuscript had been substantially revised.
However, the most important bias observed in this trial of the treatment-line, has bever been discussed in the Discussion part, including the pembrolizumab for >50% PD-L1 expression in the first-line setings.
I recommend that these points should be mentioned in the Discussion section, or the publisher.
Comments on the Quality of English LanguageI think that the manuscript had been substantially revised.
However, the most important bias observed in this trial of the treatment-line, has bever been discussed in the Discussion part, including the pembrolizumab for >50% PD-L1 expression in the first-line setings.
I recommend that these points should be mentioned in the Discussion section, or the publisher.
Author Response
Thank you for your remark and apologies for the omission. We included the following elements in the “Discussion” section of the last manuscript version regarding this point:
Our series provides homogeneous data, as we only included patients under anti-PD-1/PD-L1 monotherapy, thus increasing the relevance of biological hypotheses in secondary resistance. In addition, our cohort consists mostly of Caucasian patients with data that might biologically better apply to western countries population. For instance, one of the largest cohort of secondary resistance in aNSCLC (n=51 patients under an-ti-PD-1/PD-L1 monotherapy) originates from Korea and comprises a high proportion of EGFR-mutated NSCLC (≥17.4%).[23]
On the other hand, some heterogeneity in our cohort could arise from the inclusion of both aNSCLC receiving anti-PD-1/PD-L1 monotherapy as a first line of treatment (in case of ≥50% PD-L1 expression) and already pre-treated patients. This could also be a potential confounder with regard to the association between the type of resistance and PPS. Indeed, trials such as the KEYNOTE-024 suggest better outcomes of frontline immunotherapy, rather than as a second (or later) line, though it is not clear if this factor also influences the PPS in addition of extending the PFS on immunotherapy.[5] More generally, the number of previous lines of treatment is a classical prognosis factor in aNSCLC. The size of our sample did not allow to efficiently compare both types of resistance within each stratum of ICI line. For these reasons, we incorporated this variable in the Cox multivariable PPS model (Fig.3), which still resulted in a strong association between PPS and the type of resistance while adjusting for the immunotherapy line.